# Structural and Kinetic Views of Molecular Chaperones in Multidomain Protein Folding

**DOI:** 10.3390/ijms23052485

**Published:** 2022-02-24

**Authors:** Soichiro Kawagoe, Koichiro Ishimori, Tomohide Saio

**Affiliations:** 1Graduate School of Chemical Sciences and Engineering, Hokkaido University, Sapporo 060-8628, Hokkaido, Japan; kawagoe.soichiro.w3@elms.hokudai.ac.jp; 2Graduate School of Medical Sciences, Tokushima University, Tokushima 770-8503, Tokushima, Japan; 3Department of Chemistry, Faculty of Science, Hokkaido University, Sapporo 011-0021, Hokkaido, Japan; 4Division of Molecular Life Science, Institute of Advanced Medical Sciences, Tokushima University, Tokushima 770-8503, Tokushima, Japan; 5Fujii Memorial Institute of Medical Sciences, Institute of Advanced Medical Sciences, Tokushima University, Tokushima 770-8503, Tokushima, Japan

**Keywords:** molecular chaperone, biophysical method, multidomain protein, binding kinetics, protein structure

## Abstract

Despite recent developments in protein structure prediction, the process of the structure formation, folding, remains poorly understood. Notably, folding of multidomain proteins, which involves multiple steps of segmental folding, is one of the biggest questions in protein science. Multidomain protein folding often requires the assistance of molecular chaperones. Molecular chaperones promote or delay the folding of the client protein, but the detailed mechanisms are still unclear. This review summarizes the findings of biophysical and structural studies on the mechanism of multidomain protein folding mediated by molecular chaperones and explains how molecular chaperones recognize the client proteins and alter their folding properties. Furthermore, we introduce several recent studies that describe the concept of kinetics–activity relationships to explain the mechanism of functional diversity of molecular chaperones.

## 1. Introduction

Significant advances in protein structure prediction have led to the easy availability of protein structural information [1,2]. However, the understanding of structure formation, folding, remains one of the biggest mysteries in protein science. The newly synthesized protein searches for its own three-dimensional structure from myriad possibilities [3,4]. Despite the huge number of conformations possible, proteins often fold in seconds or even shorter periods (the Levinthal paradox) [5]. However, little is known about how proteins find efficient folding pathways and the process of structure formation. To understand the mechanism of protein folding, experimental and computational studies have been performed over several decades.

Protein folding has been traditionally studied on small single-domain proteins, including BPTI [6], ribonuclease [7], and cytochrome *c* [8,9]. These studies have provided structural insights into folding intermediates and, thus, have expanded the understanding of the major folding pathways. However, a large proportion of the proteome consists of multidomain proteins: 40–65% of prokaryotic proteins and 65–80% of eukaryotic proteins [10]. Assembling multiple domains into one polypeptide chain is believed to be an evolutionary strategy to build up a protein with novel functions [11]. Folding of larger proteins and multidomain proteins often requires highly complex processes and, therefore, takes longer. Because the intracellular environment is a crowded environment in which the concentration of macromolecules is estimated as 300–400 mg/mL [12], the risk of aggregation due to nonspecific molecular interactions exists. Therefore, the folding of multidomain proteins in the cell often proceeds with the assistance of various molecular chaperones.

Molecular chaperones can be classified based on their characteristic activity. “Foldases” assist in the folding of newly synthesized proteins. “Holdases” prevent or retard client protein folding, often for efficient protein translocation. For instance, trigger factor (TF) [13], DnaK [14], and Spy [15] work as foldases to promote client protein folding; SecB [16] and heat shock protein 40 (Hsp40) [17] work as holdases to inhibit client protein folding. The holdase activity of molecular chaperones is known to be important for protein folding, in which the chaperone slows down the folding process for higher yield [18]. Furthermore, some molecular chaperones, including caseinolytic mitochondrial matrix peptidase chaperone (Clp)/heat shock protein 100 (Hsp100) chaperones [19,20], are classified as “Unfoldases” that unwind the structure of damaged proteins [20,21]. Interestingly, chaperone activity is rather variable. One chaperone can exert multiple different activities, depending on the conditions or client proteins. For example, one chaperone can work as a foldase for one group and as a holdase for another group of client proteins. This activity switching of molecular chaperones is believed to be important in regulating the folding of multidomain proteins.

In this review, we focus on multidomain protein folding, assisted by molecular chaperones, and discuss how molecular chaperones alter the folding properties of client proteins. This review summarizes fundamental understanding and recent advances in multidomain protein folding and mechanistic insights into molecular chaperones. In particular, recent advances in structural biology in understanding how molecular chaperones recognize unfolded client proteins are discussed. Furthermore, we have introduced kinetic studies that shed light on the dynamic aspect of the chaperone–client complex. A combination of structural and kinetic information of the chaperone–client complex explains the distinct and variable activities of molecular chaperones.

## 2. Folding Process of Multidomain Proteins

Multidomain proteins often possess higher-order architecture composed of multiple structural domains that recognize each other to form a unique arrangement. The domain is sometimes “discontinuous” in which the polypeptide chain runs into another domain at the middle of the domain and then comes back to form the remaining part of the domain. Owing to this complex architecture, multidomain protein folding often requires multiple folding steps called “segmental folding” (Figure 1) [22]. Segmental folding has been studied by various techniques such as fluorescence measurement, nuclear magnetic resonance (NMR), atomic force microscopy (AFM), and hydrogen–deuterium exchange mass spectrometry (HDX-MS). In this section, we will introduce the results of AFM and HDX-MS studies.

An effective method to investigate protein folding and unfolding includes single-molecular force–extension coupled with AFM [23] or optical tweezers [24], in which the target protein is tethered to be applied for mechanical force and the unfolding process is evaluated through force applied and extension distance. The force–extension experiment has been used to investigate the unfolding of multidomain proteins, including T4 lysozyme [25], elongation factor G [26], and maltose-binding protein (MBP) [24,27,28]. MBP, one of the best studied proteins in the field of protein folding, consists of ~370 amino acids and has globular overall shape composed of N and C lobes [29]. Thus, understanding MBP folding could provide the key to understanding the folding of multidomain proteins. A study on MBP revealed that MBP consists of four structural blocks of ~100 residues [27] (Figure 2A). Although the experiment monitors the unfolding process, some structural segments of MBP were shown to have enough thermal stability, indicating that these unfolding intermediates correspond to folding intermediates [27]. The segmental folding of the multidomain protein, dihydrofolate reductase (DHFR), which is composed of two domains, has also been supported using molecular dynamics (MD) simulation studies [30]. Another simulation study on DHFR showed that if the protein has continuous and discontinuous domains, the continuous domain is preferably folded first to prevent steep entropy loss through the association of the discontinuous regions of the polypeptide chain [31].

Detection of folding intermediates is also performed using HDX experiments coupled with NMR or MS, in which target proteins unfolded in the ^2^H_2_O solvent are refolded and pulse-labeled with protons by the rapid increase in pH [8,32,33,34]. Given that solvent-exposed amide protons in the unfolded region of the protein are exchangeable at higher pH, the higher protection rate for hydrogen–deuterium exchange indicates the formation of higher-order structures. Peptide- or residue-level information using MS or NMR, respectively, is exploited to evaluate the structure of the folding intermediates. HDX-MS experiments on MBP folding also showed that MBP undergoes segmental folding [19] (Figure 2B). Interestingly, the folding units of MBP observed in the HDX-MS experiments are located in the distinct unfolding segments of MBP as seen in the AFM unfolding experiments [27] (Figure 2), which supports the idea that segments identified in the unfolding experiment represent segments can form during folding. Experiments on the slow-folding mutant of MBP show that the mutation delays the initial segmental folding, and therefore, delays the folding of the other segments, suggesting the importance of the order of segmental folding [19].

Thus, several studies agree with the idea that multidomain protein folding consists of multiple folding intermediates, with specific segments of the protein folded in an order. The importance of segmental folding in multidomain protein folding implicates possible intervention by molecular chaperones in segmental folding.

## 3. Effect of Molecular Chaperones on Protein Folding

Molecular chaperones accelerate or slow client protein folding. However, how molecular chaperones alter the folding properties of client proteins is poorly understood. Regarding foldase activity, whether molecular chaperones affect folding actively or passively is debatable. Active effect refers to the ability of the chaperone to alter the folding state or folding path of the client protein at the folding intermediate. By contrast, passive effect indicates that the chaperone indirectly affects protein folding by preventing client protein aggregation. One of the examples of passive/active arguments is seen in the case of GroEL/ES. Passive effect of GroEL/ES was shown through refolding experiments in which the refolding of the double mutant (DM) of MBP was monitored by tryptophan fluorescence [35]. At higher concentrations, DM-MBP was prone to aggregate, leading to a decrease in the apparent folding rate. In this condition, the folding rate increased by addition of GroEL/ES. However, in the absence of chloride ions, where no aggregation of MBP was detected, GroEL/ES did not increase the folding rate of DM-MBP. These observations suggest a passive-cage model in which the chaperonin does not directly affect folding but inhibits aggregation and, consequently, increases the folding yield [35]. On the other hand, an active-cage model, in which the folding rate is accelerated by confinement in a chaperonin cage, has also been proposed [36]. GroEL/ES was shown to accelerate the folding of DapA, a native client, by more than 30-fold at a very low concentration of DapA (100 pM), where the possibility of intermolecular associations, such as aggregation or assembly, can be excluded. The result supports the active effect of GroEL/ES in the folding. Folding in the restricted environment of a chaperonin cage is characterized by rapid stepwise structure formation, which effectively reduces the entropic component of the energy barrier. Approximately 30–50% of obligate GroEL clients, including DapA, share the triosephosphate isomerase (TIM)-barrel domain fold, suggesting that the physical properties of the GroEL/ES-cage are particularly suited to achieve folding catalysis for a subset of TIM-barrel domain proteins as a result of coevolution [37,38].

In addition to refolding assays, structural biology and biophysical analysis have enabled investigation of protein folding coupled with molecular chaperones at higher resolution. For instance, MBP refolding in the presence of GroEL has been examined using HDX-MS. The folding study on the MBP mutant using HDX-MS revealed that GroEL alters the process of segmental folding [19] (Figure 2B). A delay in the folding of the N-terminal segment of MBP due to mutations delays global folding, which can be partially retrieved by GroEL, indicating that the chaperone accelerates segmental folding and, therefore, global folding (Figure 1).

Single-molecular observation of protein folding in the presence of chaperones has also been reported. In force–extension experiments, the chaperone, TF, has been shown to alter the folding pathways of MBP, in which the folding intermediates of MBP can be stabilized by TF [28]. The crystal structure of TF chaperone in complex with the folded small client protein suggests the possibility of TF recognizing partially folded intermediates in addition to the unfolded protein [39]. Thus, these data suggest that the foldase activity of TF is exerted upon the formation of the folding intermediate. The effect of SecB has also been examined using force–extension experiments that have shown the holdase activity of SecB, where SecB binds unfolded MBP and prevents the formation of tertiary structures [24]. The results reveal that SecB has strong holdase activity and prevents client protein folding.

As discussed above, structural and biophysical studies have shown that the chaperones affect the early stages of the folding process to prevent or promote the formation of the intermediate folding state with segmental folding. These studies significantly extended the understanding of the mechanism of chaperone-mediated protein folding; although, detailed mechanisms remain unclear.

## 4. Structural Features of Molecular Chaperones

In the following sections, we have discussed how molecular chaperones act on the folding process of client proteins based on the structural features of molecular chaperones alone. First, we have introduced the structure of molecular chaperones. X-ray crystal structures and cryo-EM structures reveal that molecular chaperones have distinct and characteristic shapes. For instance, chaperones classified as chaperonin, including GroEL in prokaryotes [40] (Figure 3A) and tailless complex polypeptide 1 ring complex (TRiC) (also called chaperonin containing tailless complex polypeptide 1 (CCT) in eukaryotes [41] (Figure 3B), are chamber-shaped; heat shock protein 90 (Hsp90) [42] (Figure 3C) and Hsp40 [43] are V-shaped (Figure 3D); SecB [44] forms a disc-shaped tetramer (Figure 3E); and TF [45] has elongated shape that is often represented as dragon-shape (Figure 3F). The structural information of chaperones provides insights into their mechanism, summarized later.

The chamber shape of the chaperonin enables the isolation of the client protein in the chamber to prevent aggregation, and thus promote the spontaneous folding of the client protein. Furthermore, the chaperonin promotes client protein folding by changing the environment in the chamber coupled with adenosine triphosphate (ATP) hydrolysis [46]. For instance, the hydrophobic surfaces inside the chamber of GroEL are exposed and occluded coupled with ATP binding and hydrolysis [47], which can regulate the binding and release of the client protein. In the hetero-oligomeric TRiC, which consists of two stacked octamers of eight different subunits [46,48], each hemisphere of the TRiC octamer ring hydrolyzes ATP at a different rate, resulting in heterogeneous conformational changes that create a complex internal environment in terms of charge and hydrophobicity. The variation in the internal environment is believed to play an important role in the folding of multidomain proteins with diverse sequences. A correlation between the size of the proteome and the diversity of TRiC subunits exists, suggesting that chaperonin complexity is functionally optimized for the complexity of the proteome, and that the folding machinery contributes to the determination of proteome size [49]. ATP-dependent conformational change is also known for Hsp90 [50,51] and heat shock protein 70 (Hsp70) [52,53] and is coupled with binding and release of the client protein, which is considered to be important for foldase activity.

As seen above, the nucleotide-dependent conformational change in chaperones revealed by structural studies provided implications for understanding the mechanism. On the other hand, ATP-independent molecular chaperones, such as TF and Spy, can also work as foldases, and SecB can work as a holdase. However, the mechanisms for the activities of these ATP-independent chaperones are not explained by the structures of the chaperones alone.

## 5. Structural Studies of Chaperone–Client Complexes

Structural analyses of chaperone–client complexes using NMR have revealed mechanisms of client protein recognition by molecular chaperones. NMR structures of chaperones in complex with unfolded client proteins have been reported for Hsp40 [17] (Figure 3D), SecB [16] (Figure 3E), and TF [54] (Figure 3F). In each case, unfolded alkaline phosphatase (PhoA) was used as the unfolded client protein. All three chaperones recognize multiple hydrophobic regions of the client protein and maintain the substrate protein in a denatured state. For instance, TF recognizes ~130 of 471 residues of PhoA in the unfolded state [54]. SecB and Hsp40 also recognize similar regions of PhoA. These results suggest that the recognition of hydrophobic regions is a common mechanism among chaperones. The interactions among these hydrophobic regions are the driving force for protein folding (Figure 2C); however, hydrophobic regions can cause nonspecific intermolecular interactions that drive aggregation. Recognition of hydrophobic regions of the unfolded client protein enables the molecular chaperones to prevent aggregation and alter the folding process of client proteins. Thus, understanding client protein recognition elucidates how molecular chaperones act on immature client proteins.

In addition to PhoA, unfolded MBP was also used as a client protein and regions recognized by Hsp40 [17], TF [55], and SecB [16], and were determined using NMR (Figure 2D–F). As seen for PhoA, the chaperones recognize similar regions of MBP that are mostly located on the core of the MBP native structure (Figure 2C–F). Interestingly, the regions that are recognized by the chaperones also correspond to the unfolding and folding segments identified by AFM [27] and HDX-MS [19], respectively (Figure 2A,B). This suggests that the chaperones affect the folding of these segments. Furthermore, the comparison identifies that the segments undergoing slow and intermediate folding (Figure 2B) are mostly recognized by the chaperones (Figure 2D–F), whereas those undergoing fast folding (Figure 2B) are not (Figure 2D–F). This suggests a preference of molecular chaperones toward segments with higher hydrophobicity and slower folding rate in the client protein. Further studies focusing not only on the amino acid sequence of the client protein, but also on the speed of segmental folding, are needed to support this hypothesis.

In contrast to TF, SecB, and Hsp40, which recognize multiple hydrophobic stretches of client proteins, Hsp70 and DnaK recognize narrower regions of client proteins [56]. DnaK, a member of the Hsp70 family in *E. coli*, recognizes ~700 client proteins, including multidomain proteins [57]. Although several important biochemical and structural studies have been reported for Hsp70/DnaK [52,53,58,59], an important NMR study, using DNA-binding domain of human telomere repeat-binding factor 1 (hTRF1) as client protein, has reported that hTRF1 exists in equilibrium between unfolded and folded states in solution [60]. NMR data showed that the conformational landscape of hTRF1 increased upon interaction with DnaK. This can be because DnaK binding to multiple small regions of hTRF1 generates conformational heterogeneity in the bound ensemble that can be the starting point for folding upon release from Hsp70. Thus, a model for Hsp70-mediated conformational bias was proposed, in which the conformational space sampled expands as the substrate is folded [60].

By contrast, compaction of the conformational space of the client protein during folding is reported for the chaperone Spy [61]. Conformational states of unfolded mutants of Im7 (Im7U) were examined by paramagnetic relaxation enhancement measurements using nitroxide spin labeling, which provide distance information within the range of ~20Å. The data show that Im7U assumes elongated conformations in solution and more compact conformations upon the binding to Spy. The compact conformation of the substrate protein is advantageous in terms of interaction energy, and Spy may facilitate client protein folding by promoting intramolecular interactions. Visualization of the folding intermediate recognized by the molecular chaperone has been performed using a combination of MD simulation and X-ray crystallography [62]. Site-specific labeling with iodine and its anomalous scattering were used to position iodine-labeled residues of Im7. Then experimental data were used to select minimal ensemble structures from MD simulation. The results revealed that the conformations of client protein Im7 assumed unfolded, partially folded, and native-like states, suggesting that Spy enables the client protein to explore its folding landscape while being bound to the chaperone [62].

As described above, recent studies have suggested that molecular chaperones not only prevent aggregation of client proteins, but also assist folding by actively altering the conformational space and folding path. Furthermore, detailed structural studies on multiple chaperones uncover the diversity in the mechanism of folding assistance. Some chaperones eliminate the conformational space of the client protein, whereas others expand it. This diversity in mechanism can be a key to describing the functional properties of molecular chaperones.

## 6. Kinetic Studies for Molecular Chaperones and Client Proteins

As seen in the previous sections, structural studies of molecular chaperones as well as chaperone–client complexes have extended the understanding of how the chaperone alters client protein folding. By contrast, the target of the high-resolution structural study should be usually in equilibrium, which sometimes limits the understanding of the highly transient and dynamic chaperone-mediated protein folding. For understanding the process better, kinetic information is important. Recent studies focusing on the binding kinetics between the chaperone and client proteins [13,15,16,17,63,64] have revealed that the chaperone–client complex is dynamic and mediated by fast binding–release exchange [16,54]. Furthermore, the foldase and holdase activities of molecular chaperones are related to the binding kinetics between the chaperone and client protein and to the folding speed of the client protein [16]. In this study, the binding kinetics of an unfolded client protein were monitored for two chaperones: the foldase chaperone TF and the holdase chaperone SecB. A faster association rate to the client protein was found to be a key feature for stronger holdase activity [16] (Figure 4). The folding speed of the client protein has also been shown to affect foldase/holdase activity. For instance, TF acts as a holdase for client proteins with slower folding rates. As seen in chaperone–client complex structures, both TF and SecB keep the client protein in an unfolded, extended state, indicating that TF and SecB do not allow client protein folding in the complex, whereas the client protein can spontaneously fold upon release from the chaperone. When the binding kinetics are fast enough compared to the spontaneous folding rate of the client protein, the chaperone recaptures the client protein, thereby preventing folding. Thus, foldase/holdase activity is determined by the balance between chaperone–client binding kinetics and the folding rate of the client protein (Figure 4). This kinetics–activity relationship is consistent with the kinetic partitioning theory that was proposed 30 years ago and explained the mechanism of substrate selection during protein transport by SecB [65,66].

More recently, the kinetics–activity relationship has been supported by a study on TF monomer and dimer, and it has been proposed that kinetic modulation of chaperones can be coupled with the oligomerization of molecular chaperones [13,67]. Although further investigation is needed, it has been proposed that the assembly of client-binding sites after molecular chaperone oligomerization increases the association rate between the client protein and chaperone [13].

The kinetic views of chaperone-guided protein folding are important for understanding the folding of multidomain proteins. An important example is firefly luciferase, a ~60 kDa multidomain protein consisting of a large N-terminal domain and a small C-terminal domain, as a client protein for the DnaKJE chaperone system [68]. Data from fluorescence assays and HDX-MS analysis have shown that the C-terminal domain, with the faster folding rate, is rapidly released from DnaKJE, whereas the N-terminal domain, with the slower folding rate, stays on DnaKJE longer, and folding progresses upon chaperone release. Thus, DnaK recognizes the “kinetically trapped” slow-folding regions of the client protein and biases the local folding pathway, which promotes overall folding.

## 7. Catalytic Domains in Molecular Chaperones

The catalytic domain of molecular chaperones modulates the folding speed of the client protein, thereby regulating the activity–kinetics relationship. Several molecular chaperones have catalytic domains. For example, TF, SurA, and PrsA contain the peptidyl-prolyl *cis/trans* isomerase (PPIase) domain [45,69,70,71] (Figure 3F–I) and members of the protein disulfide isomerase [6,72] family have catalytic domains for disulfide bond formation and isomerization. Furthermore, chaperones often make complexes with catalytic domain-containing cochaperones. A well-known example includes Hsp90 in complex with members of the FK506 binding protein (FKBP) family [73,74,75]. Because specific *cis/trans* conformations of peptide bonds and disulfide bonds between specific pairs of cysteine residues are required in the native fold, proline *cis/trans* isomerization and disulfide isomerization can be a rate-limiting step in protein folding. Thus, *cis/trans* and disulfide isomerization, mediated by the catalytic domains of molecular chaperones, can accelerate the speed of client protein folding and, therefore, change the balance between binding kinetics and folding speed.

The mechanism used for the isomerization of the TF chaperone by FKBP-type PPIase has been explained in detail. NMR structure analysis revealed that TF PPIase recognizes the proline-aromatic amino acid sequence and MD simulation analysis revealed stabilization of the intermediate transition state by the formation of intermolecular hydrogen bonds, which lowers the energy barrier of the transition state and catalyzes isomerization [55]. TF functions as a foldase by recognizing the proline-aromatic sequence in the hydrophobic stretch of the substrate protein, which is eventually folded into the core of the protein in the native fold. As for other types of PPIase sequence motifs, Pin1, which regulates signal transduction, recognizes phosphorylated serine/threonine-proline sequence in the loop region [76,77,78]; and cyclophilin A has a binding preference for glycine-proline sequence. [79]. Although the substrate recognition mechanism of PPIases has not been fully elucidated, they are believed to promote substrate folding by preferentially catalyzing proline isomerization in regions important for folding.

## 8. Conclusions

In this review, we summarized our current understanding of molecular chaperone-mediated protein folding. The main targets in the study of protein folding used to be small proteins, but more recently larger proteins and multidomain proteins, including MBP and lysozyme, are being studied. The folding of large proteins and multidomain proteins consists of multiple segmental folding steps and the regulation of the timing and order of the folding steps can be critical for efficient folding. Molecular chaperones can alter the folding paths by recognizing client proteins in unfolded or folded intermediate states. Advances in structural biology, biophysics, and computational science have made detailed views of protein-folding intermediates and chaperone–client complexes available, and, consequently, the mechanistic understanding of chaperone-mediated protein folding has grown. Studies on the structure, dynamics, and kinetics of chaperone–client complexes are warranted in the future for understanding protein folding at atomic resolution. Given that the protein folding is also a considerable issue in the medicinal studies on protein-misfolding diseases or recombinant protein production technologies for protein drugs and industrial enzymes, understanding protein folding will be of great importance in several research and industrial fields.

## Figures and Tables

**Figure 1 ijms-23-02485-f001:**
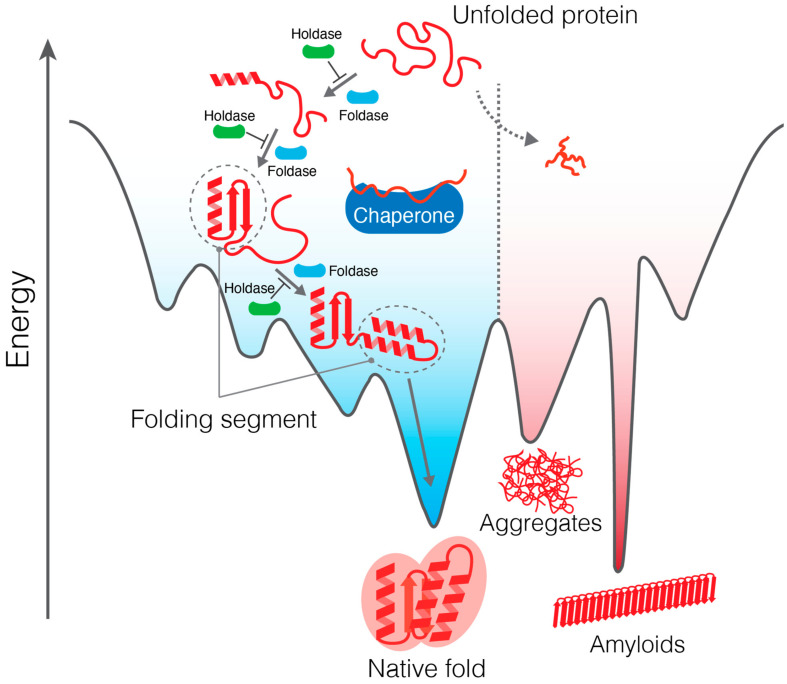
Protein folding landscape and folding paths. The unfolded protein proceeds through multiple folding steps in which a segment of the protein is folded in an order. The chaperones assist in segmental folding and prevent aggregation.

**Figure 2 ijms-23-02485-f002:**
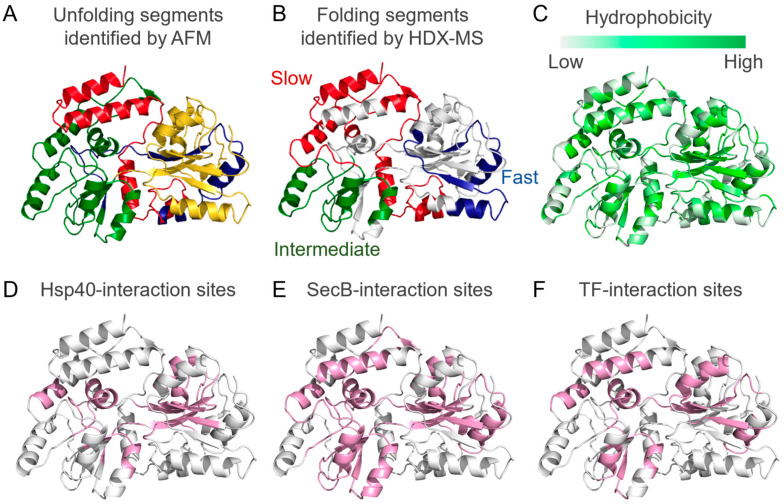
Comparison between folding segments and chaperone-recognition sites of maltose-binding protein (MBP): (**A**) The unfolding segments of MBP identified by atomic force microscopy (AFM) unfolding experiments [27] are shown in yellow (1–113), green (114–243), blue (244–295), and red (296–366). (**B**) The folding segments of MBP identified by the hydrogen–deuterium exchange (HDX)-MS refolding experiment [19]. The segments with fast (22–44 and 264–279), slow (161–209, 290–339, and 352–370), and intermediate (116–150 and 210–235) folding rates are shown in blue, red, and green, respectively. (**C**) Mapping of the hydrophobicity on the structure of MBP. (**D**–**F**) Regions recognized by Hsp40 [17] (8–14, 55–71, 89–114, 148–162, 193–200, 225–233, 244–252, and 259–266) (**D**), SecB [16] (6–12, 59–71, 84–121, 145–181, 194–285, and 339–351) (**E**), and TF [31] (6–11, 42–52, 61–65, 149–161, 194–201, 213–219, 225–233, 239–247, 258–270, 277–285, and 340–350) (**F**) are shown in pink.

**Figure 3 ijms-23-02485-f003:**
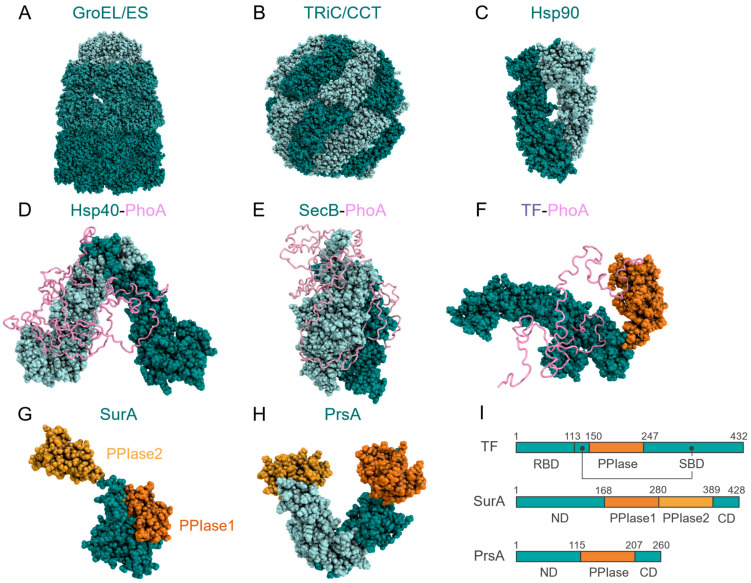
Three-dimensional structures and domain organizations of molecular chaperones. (**A**–**C**) Three-dimensional structures of GroEL/ES (PDBID 1PCQ) (**A**), TRiC/CCT (PDBID 7LUM) (**B**), and Hsp90 (PDBID 2CG9) (**C**). (**D**–**F**) Structures of Hsp40 (PDBID 6PSI) (**D**), SecB (PDBID 5JTL) (**E**), and TF (PDBID 2MLX) (**F**) in complex with the unfolded client protein PhoA. The PPIase domain of the TF is shown in orange. (**G**,**H**) Three-dimensional structures of SurA (PDBID 1M5Y) (**G**) and PrsA (PDBID 4WO7) (**H**). The PPIase domains are shown in orange. (**I**) Domain organizations of molecular chaperones having PPIase domain. The abbreviations used are: RBD, ribosome binding domain; SBD, substrate binding domain; ND, N-terminal domain; CD, C-terminal domain.

**Figure 4 ijms-23-02485-f004:**
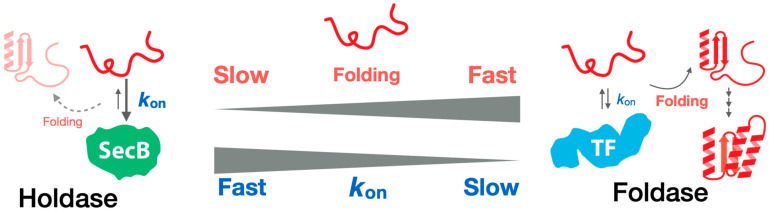
Activity–kinetics relationship seen in chaperone-mediated protein folding. The foldase/holdase activity of a chaperone can be explained by the balance between the chaperone–client binding kinetics and the folding rate of the client protein.

## Data Availability

Not applicable.

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
