# Peer review of "Structural and Kinetic Views of Molecular Chaperones in Multidomain Protein Folding"

_ijms, 2022, doi:10.3390/ijms23052485_

Round 1
Reviewer 1 Report
Soichiro et al. has presented a thorough review on the multidomain protein folding by molecular chaperones. The authors have focused on the structural studies of molecular chaperones alone and in complex with the client proteins. They have also presented a kinetic view of molecular chaperones in folding process. Overall, the study is of great interest and for further improvement of the article I have few suggestions or comments to the authors.
Comments:
- Line 35 remove how in the sentence it has been written twice
- Line 47 specify the concentration is of protein or some other biomolecule
- Line 54 write full form of DnaK and Spy.
- Line 55 write full form of SecB and Hsp40.
- Line 58 write full form of Clp/Hsp100.
- Section 4 the title "structural studies of molecular chaperones" can be changed as there is no structural study of molecular chaperone.Instead the mechanism of molecular chaperones in client protein folding is explained in this section
- Line 188 and 189 can be removed once renaming of section is done
- In section 4 some figures can be provided for the chaperones used as an example
- In figure 2 the amino acid sequence of the MBP with highlighted hydrophobic residues can be placed for better understanding of section 5
- Provide figure for section 5 for better understanding of the complex
- For section 7 include figures for domain organization, it help in understanding.
Reviewer 2 Report
A manuscript entitled „Structural and Kinetic Views of Molecular Chaperones in Mul-2 tidomain Protein Folding “was submitted by Kawagoe et al. for publication in International Journal of Molecular Sciences. In this review manuscript the authors describe the role of chaperones on the folding of proteins. The possible mechanisms of structural changes are described. It is followed by description of the kinetics of the protein folding processes. Some examples are used to illustrate the process of protein folding. The article is informative and well written fulfilling the requirements of a review. I miss in this article a compact summary on the methodological approaches used to study protein folding processes involving chaperons. It would significantly improve the quality of the manuscript and resulted in a broader interest in the manuscript. A minor review is recommended.
